# Structural Biology of Nanobodies against the Spike Protein of SARS-CoV-2

**DOI:** 10.3390/v13112214

**Published:** 2021-11-03

**Authors:** Qilong Tang, Raymond J. Owens, James H. Naismith

**Affiliations:** 1Structural Biology, The Rosalind Franklin Institute, Harwell Science & Innovation Campus, Didcot OX11 0FA, UK; qilong.tang@pmb.ox.ac.uk; 2The Wellcome Centre for Human Genetics, Division of Structural Biology, University of Oxford, Headington, Oxford OX3 7BN, UK

**Keywords:** COVID-19, antiviral therapy, protein complexes, single-chain antibodies

## Abstract

Nanobodies are 130 amino acid single-domain antibodies (V_H_H) derived from the unique heavy-chain-only subclass of Camelid immunogloblins. Their small molecular size, facile expression, high affinity and stability have combined to make them unique targeting reagents with numerous applications in the biomedical sciences. The first nanobody agent has now entered the clinic as a treatment against a blood disorder. The spread of the SARS-CoV-2 virus has seen the global scientific endeavour work to accelerate the development of technologies to try to defeat a pandemic that has now killed over four million people. In a remarkably short period of time, multiple studies have reported nanobodies directed against the viral Spike protein. Several agents have been tested in culture and demonstrate potent neutralisation of the virus or pseudovirus. A few agents have completed animal trials with very encouraging results showing their potential for treating infection. Here, we discuss the structural features that guide the nanobody recognition of the receptor binding domain of the Spike protein of SARS-CoV-2.

## 1. Introduction

Severe acute respiratory syndrome coronavirus 2 (SARS-CoV-2) is the seventh coronavirus that has infected humans [1] and, like SARS-CoV and MERS-CoV, infection can result in severe diseases [1]. SARS-CoV-2 has been uniquely dangerous, because it spreads extremely efficiently in the naïve human population. In the period from 1 March 2020 to 6 July 2021, around four million people have been confirmed to have died with COVID-19 (www.ourworldindata.org, accessed on 2 November 2021). The arrival of safe and effective vaccines [2,3,4,5] has prevented even higher death tolls. To date, no vaccine has proven 100% effective and there are concerns about waning immunity/escape mutants [6]. Thus, an effective antiviral treatment is highly desirable. Several therapies based on human monoclonal antibodies have been approved for use; a recent overview of progress in this field has been published [7]. These molecules work by binding to the Spike protein of the virus and preventing it from engaging with the angiotensin-converting enzyme 2 (ACE2) on the surface of human cell in the first step of infection. The rapid deployment from the lab to clinic reflects the accumulated knowledge gained from well-established use of antibodies in medicine, where they have revolutionised treatments ranging from cancer to arthritis (recently reviewed in [8]). From a practical standpoint, such therapies directed against COVID-19 may be limited to relatively rich countries with advanced healthcare systems, since production needs specialized cell culture, and administration of these agents requires injection or infusion [9].

Human antibodies are comprised of two protein chains, the heavy and the light chain (Figure 1a). The epitope binding region (paratope) is spread across the variable domains of both the light and heavy chains. In addition to the conventional two-chain antibody, camelids produce a single (heavy)-chain variant, that is, without a light chain. The paratope of this heavy-chain-only antibody is confined to the approximately 130-residue-variable domain (V_H_H). Nanobodies share a common fold, two antiparallel β-sheets (Figure 1b) with secondary structure elements connected by loops and turns (Figure 1c). Three complementarity determining regions (CDR1, 2 and 3) are varied to create the epitope recognition site (Figure 1d). The V_H_H domain is usually referred to as a nanobody [10] (Figure 1b–d). Traditionally, nanobodies were derived from immunisation of llama with a small amount of purified protein injected into the animal [11]. More recently, animal-free methods of nanobody generation have been reported. These laboratory methods commonly start from screening a commercial naïve library to identify binders [12]. The “hits” from the library are typically low affinity (K_D_ in the nM to μM range) but are then optimised by directed PCR mutagenesis [13], ribosomal display [14], CDR randomisation [15] and error prone PCR [16,17] followed by selection, in a process that mimics in vivo maturation (for a review, see [18]). There are alternatives to library screening, and these include grafting on CDRs from conventional antibodies followed by optimisation [19]. The compact single domain of the nanobody is straightforward to engineer and can be heterologously expressed in bacteria, yeast and human cells. The simplicity of the manipulation and production of the molecules is a particular advantage over conventional two-chain antibodies; thus, nanobodies have become an extremely valuable reagent in biochemistry [20], structural biology [20] imaging [21] and diagnostic assay development [22]. A sandwich ELISA for the detection and quantitation of Spike protein using nanobodies has recently been reported [23]. The small size and stability of nanobodies means that they have advantages over whole antibodies in terms of greater tissue penetration and the potential as inhaled biotherapeutics. Nanobodies against respiratory syncytial virus have been successfully administered directly to the airways, [24] thus avoiding treatment by injection/infusion. For therapeutic applications, nanobodies are often engineered into oligomeric forms to harness avidity. There are several strategies for oligomerisation, including fusion to Fc [25], head-to-tail fusions of nanobodies [26] or fusion to other oligomeric proteins [27]. The drug Caplacizumab [28], a bivalent nanobody, has now been approved by the FDA for treatment of thrombotic thrombocytopenic purpura. Other nanobodies are moving through trials as therapeutics [29] and diagnostics [30]. Studies have already shown that highly potent anti-COVID-19 nanobodies can be administered by injection and by inhalation with potent therapeutic effect.

## 2. Discussion

### 2.1. Cluster 1 (or Class 4)

Within months of the sequence of the SARS-CoV-2 virus being identified and sequenced, the first reports of nanobodies against COVID-19 emerged. A search of PubMed with the terms “nanobody” and “covid” returned over 50 papers. The first to be structurally characterised was named V_H_H72 [31], which had been raised previously to the Spike protein of SARS-CoV-1. This nanobody also bound to the receptor-binding domain of the Spike of SARS-CoV-2 (Figure 2a), though the structure with the RBD from SARS-CoV-1, not SARS-CoV-2, was reported. The V_H_H72 nanobody does not bind to the region of RBD that binds to the ACE2 receptor, but rather, “on the side”, at a location which is conserved between SARS-CoV-1 and SARS-CoV-2 (Figure 2a). V_H_H72 binds to the RBD molecule in a manner which competes with ACE2 binding due to steric clashes (Figure 2a). Although the result is the same, the virus does not engage its receptor and is thus neutralised; it is a different mechanism from direct competition with ACE2 for the same epitope. A study of human antibodies designated agents that bind to this epitope as cluster 1 [32], and a similar study designated this epitope as class 4 [33]; no definitive naming convention has emerged with others known [34,35,36]. Subsequent to theV_H_H72 (RCSB 6WAQ) study [31], other nanobodies that bind to cluster 1 have been reported, these include C1 (RCSB 7OAU, 7OAQ, 7AOP) [37], F2 (7OAY) [37] (Figure 2b), Sb68 (an entirely synthetic protein) (7KLW) [38], V_H_H-U (7KN5) [39], V_H_H-W (7KN7) [39], NM1226 (7NKT) [40], Nb12 (7MY3) [41], Nb30 (7MY2) [41] and WNb10 (7LX5) [42]. The region of the RBD surface in contact with the nanobody differs between the structures (Figure 2c); however, in all complexes, there is a common region on RBD, a β-strand (residues S375 to Y380). The cluster 1 nanobodies use different elements of their structure to make contact with this β-strand. In V_H_H72, C1, WNb10 and Sb68 the interaction resembles that of an anti-parallel β-strand to β-strand contact, primarily using CDR3 of the nanobody (Figure 2a). Consequently, the V_H_H72, C1 and Sb68 complexes with RBD appear as two molecules arranged side by side (Figure 2a). F2, V_H_H-U and V_H_H-W use CDR3 but bind with a more “end-on” arrangement (Figure 2b). The nanobody SR31 (7D2Z) [43] binds to this face but does not utilise the β-strand. In fact, this nanobody binds an α-helix G532 Q539 that sits slightly outside RBD (Figure 2d). Superimposing the ACE2 complex [44] reveals that the orientation of some of the nanobodies would, like V_H_H72, prevent the binding of ACE2, whilst others, like F2, do not prevent ACE2 binding but still neutralise the virus [37]. As of yet, we have insufficient data to determine whether preventing ACE2 binding correlates with increased neutralisation potency.

Analysis of the affinity, the surface area of the RBD buried and the number of hydrogen bonds in the complexes are listed in Table 1. The nanobodies show three orders of magnitude difference in affinity; this is remarkable variation that speaks to the potential for rational engineering to improve affinity. However, analysis of buried surface area and H-bonds shows no discernible trend or correlation with affinity. Clearly, new tools are needed to correlate affinity with structure before such rational redesign becomes possible. However, the demonstration of pM potency shows that some cluster 1 binding nanobodies have potential as broad spectrum virus neutralisers.

RBD shows only infrequent mutations of residues S375 to Y380 [46,47], indicating that cluster 1 nanobodies will be cross-reactive to a wide range of strains. However, mutations at K378 are known, but their effect on the binding of the nanobodies is not described [46]. Repeated passage of the virus has generated escape mutations (Y369H, S371P, F377L and K378Q/N) [39], but these are not currently circulating.

### 2.2. Cluster 2 (or Class 1 and 2)

The first nanobody reported to recognise the cluster 2 epitope was H11-H4. This nanobody derives from a parent H11 class of nanobodies with affinities varying from around 1 μM to 10 nM [51] (Figure 3a). The nanobody competes with ACE2 for binding to the RBD [51] (Figure 3A). Many further examples of this class have followed, including C5 (7OAO) [37], H3 (7OAP) [37], mNb6 (7KKL) [52], Nb6 (7KKK) [52], Nb20 (7JVB) [53], NM1230 (7B27) [40], Sb23 (7A25) [54], Ty1 (6ZXN) [55], Nanosota-1 (7KM5) [56], VHH-E (7B14) [39], WNb2 (7LX5, 7LDJ) [42], MR17 (7C8W) [57], SR4 (7C8V) [57], Sb14 (7MFU), Sb16 (7KGK) [38] and Sb45 (7KLW) [38]. A summary of the interactions between the nanobody and RBD is given in Table 2. These nanobodies sit on the top face of the RBD molecule, and although it is possible to find pairs of molecules that have very little overlap in surface area, as a group they form an overlapping continuum of surface residues that covers the strand P491 to T500 and G446 to Y449 (Figure 3b). The residue Q493 is found in contact with every nanobody and makes important contacts with ACE2 [44]. The observed competitive binding of class 2 molecules with ACE2 is rationalised by this overlap and that of other residues (Y489 toY505) [44] (Figure 3b). Interestingly, superposition of the collection of nanobodies (Figure 3b) engages a surface area comparable to that spanned by both chains of an antibody (Figure 3c).

The beta variant [6] of SARS-CoV-2 carries amongst other changes a mutation at E484. This change has led to concerns about immune evasion as this mutation leads to a notable deterioration in the human antibody response (whether by vaccine or by previous infection) [59]. The very first nanobody structure noted the central nature of a salt bridge between E484 of RBD and R52 of CDR2 of H11-H4. The salt bridge immobilises the Arg which, in turn, makes a π-cation interaction with F490 of RBD [51] (Figure 3d). The H11-H4 nanobody was derived from a naïve library hit (nanobody H11) which produced a series of daughter molecules with changes in a 7-residue stretch in CDR3 [51]. Sequence analysis suggested that it is this clamping of R52 by E484 and F490 that drives the affinity of the H11 class of nanobodies. The nanobody H3 also preserves this interaction with R52 of CDR2, but otherwise binds distinctively to the H11 class [37] (Figure 3d). Two nanobodies (C5, H3) which rely on the E484 F490 π-cation interaction have been reported to no longer bind to RBD with the E484K mutation (beta variant) [37].

Nb6 utilising R54 of CDR2 makes a similar interaction but with less ideal geometry in the salt bridge. The EM structure of mNb6 with Spike does not show this interaction, but it is clearly possible by adjusting side-chain conformations, and its absence may reflect the lower resolution of the EM structure [52]. Despite having very little overlap in their binding surface, the C5 molecule captures the same interaction using R31 of CDR1, although the Arg residue enters the clamp from the opposite site [37]. Nb20 [53] also uses R31 from CDR1 to make the same interaction but with little other overlap. Interestingly, MR17, which has the potential to make this contact with R59, does not; however, R33 makes contacts with the backbone carbonyls of 486 and 488 of RBD and π-cation interactions with Y489 [56]. Other nanobodies make salt-bridge or hydrogens bonds with the E484. It is noticeable that the sub nM binders characterised to date utilise the salt-bridge π-cation clamp; thus, it appears that irrespective of whether the nanobody is raised in llama or isolated by library screen with DNA shuffling, this interaction is particularly attractive in gaining affinity. The decrease in the human antibody response to the beta strain might suggest that the human immune system likewise utilises this clamp interaction. In fact, virus escape maps for polyclonal antibody responses to infection resemble a single monoclonal antibody class that is vulnerable to mutations at E484 [46].

### 2.3. Spike Binding and EM Conformational Analysis

In the lipid-bound form of the Spike protein, all three RBDs are found in the three-down configuration [60] (Figure 4a); the cluster 1 epitope is inaccessible to nanobodies and antibodies. C1 and F2 were reported to disrupt the trimeric structure of the Spike [37]. The cluster 1 nanobodies (Nb30) (7MY2) [41] (Figure 4b) and V_H_H-V (in combination with the cluster 2 V_H_H-E nanobody) (7B18) [39] (Figure 4c) are found in a 3-up arrangement when bound to the Spike. Nb12 (7MY3) [41] is found as 2-up 1-down Spike (Figure 4d).

Unlike the cluster 1 nanobodies, the surface engaged by the cluster 2 is exposed in the locked-down form of the Spike. The Sb23 nanobody complex with Spike is found as the *2-up 1-down* configuration (Figure 4e). The nanobodies H11-H4, Ty1 and Sb45 are found bound to Spike with a *2-down 1-up* arrangement (Figure 4f). In addition, the nanobody Sb45:Spike complex is observed in both *2-up 1-down* and *1-up 2-down* arrangements [38]. The C5 nanobody (Figure 4g) Nb6, and mNB6 Spike complexes are found bound to the *all-down* RBD arrangement. In some cases, modelling suggests that the nanobody would clash with other arrangements of the RBD, whilst in others, it appears that the presence of the nanobody has introduced additional contacts that favour a particular arrangement. This behaviour reveals one of the other uses of nanobodies in structural biology to trap different conformations of proteins that are relevant to their function.

### 2.4. The Cluster 1 and Cluster 2 Epitopes

The core of the cluster 1 epitope is the β-strand of RBD (residues S375 to Y380), whilst the core of the cluster 2 epitope consists of two regions of structure (G446 to Y449 and P491 to T500) (Figure 5a). These epitopes represent two spatially distinct surface patches on RBD (Figure 5b). The location means it is possible to bind two molecules. This was demonstrated to give rise to additive neutralisation when the antibody CR3022 and H11-H4 were combined [51]. The EM structure of cluster 1 V_H_H-V and cluster 2 V_H_H-E nanobodies [39] (Figure 4c) is an example of this property. The crystal structures of F2 (cluster 1) and H3 (cluster 2) have been described [37], and an ELISA assay for Spike protein relies on the ability of C5 and C1 to both bind at the same time to RBD (Figure 5c) [23].

### 2.5. Comparison with Human Antibodies

The SARS-CoV-2 binding epitopes of human monoclonal antibodies isolated from the B cells of either convalescent patients or following vaccination have been studied in detail by cryo-EM and X-ray crystallography [33,34,61]. The picture that emerges is similar to the pattern for nanobody binding with human antibodies broadly falling into the four classes corresponding to epitopes that map to (i) the RBD-ACE-2 interface (Cluster 2; Classes 1 and 2), exemplified by REGN 10933 [62], (ii) the outer face of the RBD exemplified by S309 (Class 3) [63] and (iv) the inner face of the RBD exemplified by CR3022, which does not compete with ACE2 for binding [64,65] (Cluster 1, Class 4) [33]. As described above, nanobodies have been identified that correspond to Class 1, 2 and 4, but so far not class 3, which are focused on the region close to the N-glycosylation site at N343. In general, Class 1 and 2 (cluster 2) human antibodies are more strongly neutralising than Class 3 or 4, reflecting direct competition for ACE-2 binding, though they are more vulnerable to escape mutations, in particular those at RBD residues 417, 484, [34,46]. The mechanism of neutralisation for cluster 1 nanobodies may be more varied; some such as V_H_H72 [31] and C1 [37] prevent ACE2 binding by creating steric clashes, whilst others such as F2 (at least in the monomeric form) [37] do not alter ACE2 binding. Rather such “non-competitive” neutralising agents may work by perturbing the trimeric Spike structure [37,66]. Antibodies that cross-react with SARS-CoV-1 and potentially other closely related bat beta-coronaviruses come from Classes 3 and 4, reflecting the higher sequence conservation in the region of the RBD recognised by these antibodies.

Both the variable domains of both heavy and light chain contribute to the paratopes of neutralising SARS-CoV-2 antibodies, which means that, in contrast to the single-domain nanobodies, the area of binding is typically much larger. Thus, some nanobodies that bind to cluster 1 do not compete with ACE2 binding. Since it has been suggested that agents that directly compete with binding of ACE2 are the most likely to promote the emergence of escape mutations [67], such cluster 1 nanobodies may be particularly attractive as anti-virals. Despite the smaller surface engaged, the binding affinity of nanobodies is at least as good as whole antibodies, and in some cases higher. This may reflect the overall smaller size of nanobodies and the relatively extended CDR3 loops that enable precise and intimate engagement with the Spike protein.

### 2.6. Biological Activity

Reflecting the urgency of finding treatments for the pandemic, several studies using neutralising nanobodies in animal trials have reported very encouraging results. In all cases, the nanobodies were multimerised to gain binding avidity, either by fusion to IgG1 Fc to construct a bivalent heavy-chain-only whole antibody or by joining end-to-end using flexible linkers of different lengths to produce dimeric or trimeric binders. As suggested for whole antibodies [34], these multivalent nanobodies may neutralise the virus by a mechanism that involves binding within a single trimer (intermolecular) or cross-linking two or three trimers (intramolecular) on the virus surface or both.

Prophylactic administration by intraperitoneal injection (IP) of WNb2-Fc (cluster 2) at 5 mg/kg 24 h before challenge showed almost complete protection against SARS-CoV-2 (hCoV-19/Australia/VIC2089/2020) [42]. A similar protective effect was found when 2.5 mg of the MR3 dimeric nanobody (no definitive structural data were recorded but it is suggested to be cluster 1) were administered IP to hamsters 6 h before challenge (approximately 100 mg/kg) with hCoV-2/BetaCoV/Munich/BavPat1/2020 [57]. IP administration of nanobody, Nanosota-1-Fc (cluster 2) at 20 mg/kg demonstrated prophylactic protection when given 24 h before challenge and therapeutic 4 h post challenge at 10 mg/kg (all IP with SARS-CoV-2 (US_WA-1 isolate) [56].

The size and stability of nanobodies makes them attractive for direct delivery to the airways by inhalation [68]. In fact, an inhalable nanobody trimer (PiN-21) raised to the RBD showed protection when administered concurrently with viral challenge and therapeutic benefit 6 h after infection (both at 0.6 mg/kg nasal) [68]. Although no structural data have been reported on the source nanobody (Nb21) for PiN-21, it can be assigned to cluster 2 on the basis of epitope binning [53]. Similarly, the C5 nanobody (cluster 2) when presented as head-to-tail trimeric fusion showed both prophylactic (2 h, 4 mg/kg) and therapeutic efficiency (0.4 mg/kg nasal (topical) 1 day post infection) against hCoV-2/human/Liverpool/REMRQ0001/2020). By 24 h, viral infection is established in the hamster model, so efficacy by treatment at this time point confirms the therapeutic potential of cluster 2 antibodies delivered by nasal installation. Combining different non-overlapping classes of nanobodies may be particularly attractive in therapy as a means of avoiding escape mutations, as demonstrated by [42]. Further, the modular nature of nanobodies makes it feasible to generate multi-paratopic versions that combine more than one binding specificity into a single molecule [39]. As well as potential therapeutics, anti-SARS-CoV-2 nanobodies have diagnostic value and the non-overlapping nature of the Class 1 and Class 2 epitopes has also been used for a sandwich ELISA assay for detection of the viral spike protein [23].

## 3. Conclusions

Nanobodies have shown themselves to be extremely potent molecules in neutralising the virus. The range of affinities is comparable to those observed for human antibodies. Encouragingly, it has been possible to use structural information to interpret the binding data, and given the emergence of variants of SARS-CoV-2, this has been particularly useful. Nanobodies have been found that bind to all known conformers of Spike protein. Modelling suggested H11-H4 could bind to different conformations of Spike [51] and the Sb23 Spike complex [54] was found in two conformations. The ability to engage Spike protein in different conformations may allow nanobodies to neutralise the virus at multiple stages in its life cycle, contributing to their potency. The nanobodies have reached advanced testing in animals with very promising results, both as a prophylactic and a therapeutic.

## Figures and Tables

**Figure 1 viruses-13-02214-f001:**
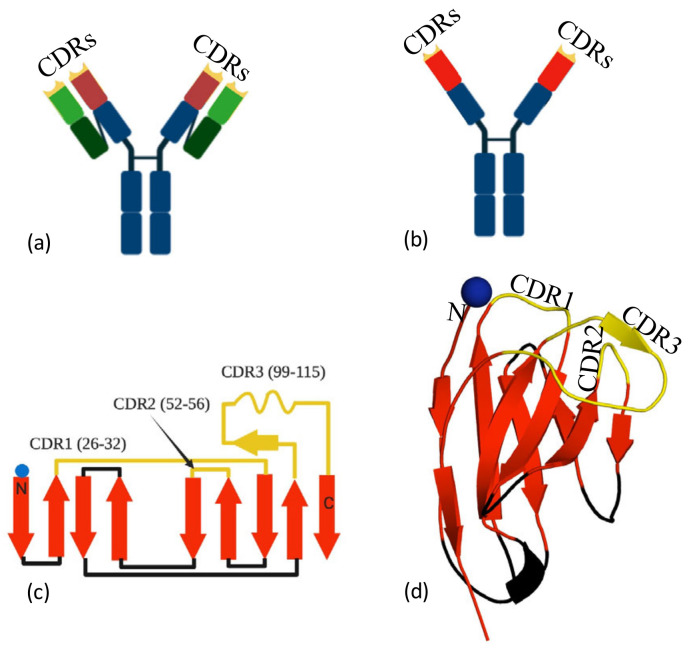
Nanobodies, (**a**) a conventional antibody, the complementarity determining regions (CDRs) (yellow) are formed by both the heavy (wine) and the light (green) chains. (**b**) Camelid antibodies only have a heavy chain; thus, the CDRs (yellow tips) are located within one domain (red). The isolated red domain is known as the nanobody. (**c**) Secondary structure of a nanobody, with approximate boundaries of the three CDRs (yellow). Sheets are colored red and interconnecting loops in black. (**d**) The three-dimensional structure of the nanobody, colored as 1c. Epitope recognition is controlled by the CDRs.

**Figure 2 viruses-13-02214-f002:**
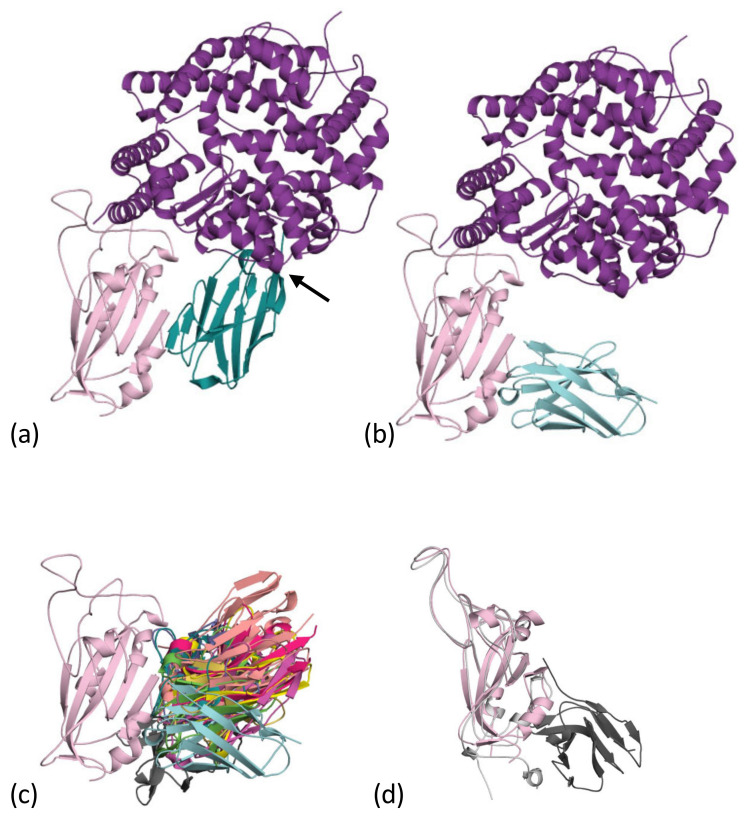
Cluster 1 nanobodies: (**a**) the first nanobody to be reported in this cluster was V_H_H-72 [31]. The nanobody (teal), binds to the “side” of the RBD of the SARS-CoV-2 (pale magenta). This is a different location than the binding site of human ACE2 (purple). V_H_H-72 blocks ACE2 binding to RBD binding through steric hinderance (arrow). We termed this “side-by-side” binding orientation. The structure is a composite of 6WAQ [31] and the ACE2 RBD complex [44,45] (6M0J). (**b**) The F2 nanobody (cyan) binds essentially to the same epitope but its orientation means, unlike V_H_H-72, that it is not competitive with ACE2. We term this an “end-on” binding orientation. The RBD is colored as 2a. The structure is a composite of 7OAY [37] and the ACE2 RBD complex [44,45] (6M0J). (**c**) A superposition of the nanobodies listed in Table 1 reveals that they all bind to the same region but adopt a range of binding orientations. The orientations allow competitive or non-competitive behavior to be rationalized. RBD is colored as 2a. (**d**) The SR31 nanobody (dark grey), which belongs to this cluster, also recognizes a portion of structure normally considered out with the RBD domain (pale pink from 7OAY). The larger RBD from the SR31 complex structure is taken from 7D2Z [43] and is colored light grey.

**Figure 3 viruses-13-02214-f003:**
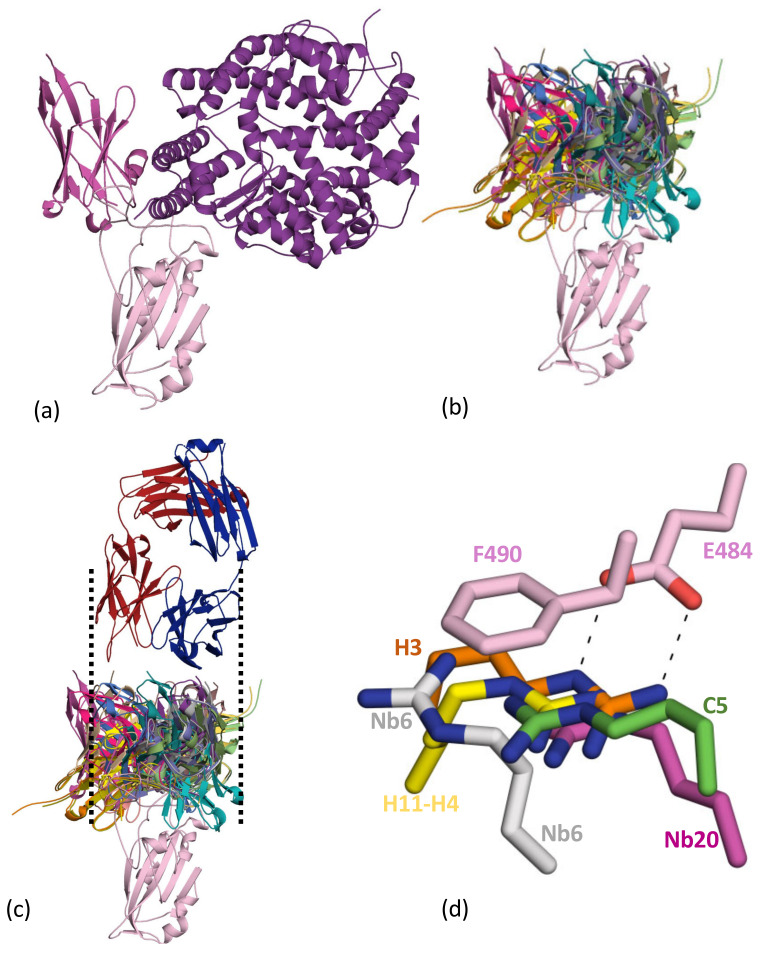
Cluster 2 nanobodies: (**a**) The first nanobody to be reported in this cluster was H11-H4 (magenta). This molecule was competitive in its binding to RBD with ACE2 (both colored as Figure 2a). The structure is a composite of 6ZBP [51] and the ACE2 RBD complex [44,45] (6M0J). (**b**) A superposition of the 18 nanobodies from Table 1 reveals they all bind to the same region of the RBD (top), and all are competitive with ACE2 binding. (**c**) The range of orientations exhibited by the nanobodies mimics the size of a human antibody (CR3022) [58] shown for comparison with lines to make the variable regions. (**d**) Arg from the nanobody makes a combined π-cation salt-bridge interaction with E484 F490 of RBD. This interaction was first seen in the H11-H4 structure [51] and is central to other cluster 2 nanobodies (H3, C5, Nb6, Nb20).

**Figure 4 viruses-13-02214-f004:**
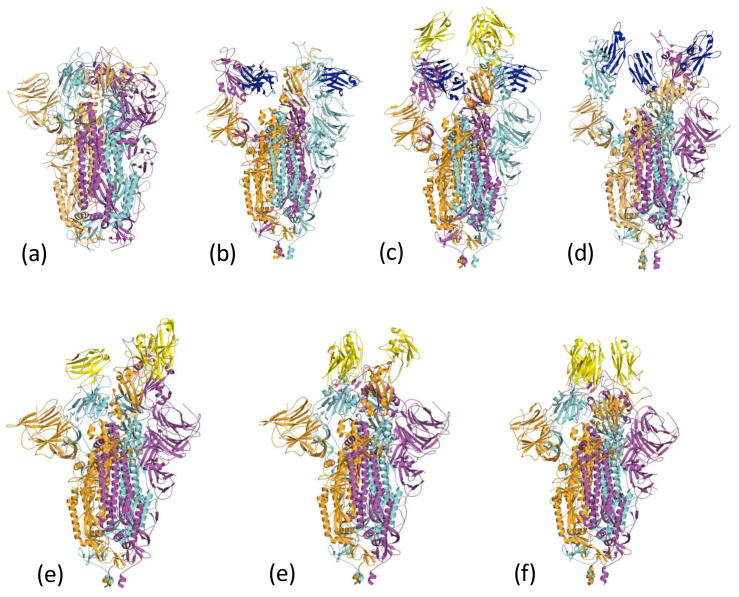
Nanobodies can influence the conformation of the spike protein. (**a**) The all-down form of the Spike [60] masks the cluster 1 epitope. The monomers of the Spike trimer are coloured orange, cyan and magenta. (**b**) Nb30 (dark blue), a cluster 1 nanobody, induces the 3-up conformation of the Spike trimer (coloured as Figure 4a). The structure is 7MY2 [41]. (**c**) The EM structure 7B18 [39] reveals a *3-up* structure of the Spike which colored as Figure 4a. In this structure, the cluster 1 nanobody V_H_H-v is colored blue, whilst the cluster 2 nanobody V_H_H -E is colored yellow. (**d**) The EM Scheme 1. Nb12 (colored blue) with Spike (colored as Figure 4a) exhibits a 2-up 1-down conformation. The structure is 7MY3 [41]. (**e**) The cluster 2 nanobody Sb23 (yellow) induces a *2-up 1-down* conformation of Spike (colored as Figure 4a). The structure is 7A29 [54]. (**f**) The cluster 2 nanobody H11-H4 colored yellow, induces a *2-down 1-up* configuration of the Spike (colored as Figure 4a). The structure is 6ZXN [51]. (**g**) The cluster 2 C5 nanobody colored yellow, induces a *3-down* configuration of the Spike (colored as Figure 4a). The structure is 7OAN [37].

**Figure 5 viruses-13-02214-f005:**
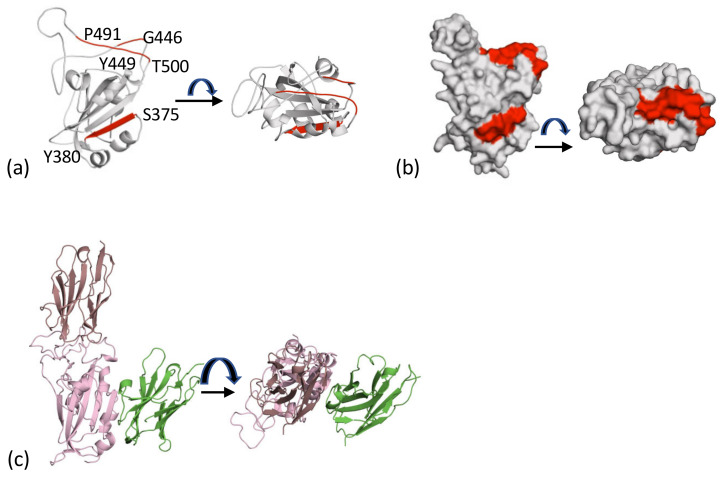
The two epitopes that drive recognition. (**a**) The cluster 1 epitope in red facing and is found on β-strand (S375 to Y380). Cluster 2 is also shown in red and is on the top of the molecule; it is formed by P491 to T500 and G446 to Y449. (**b**) The surface representation of the core of the two epitopes in Figure 5a; these are targeted by both human antibodies and llama nanobodies. (**c**) It is possible for more than one nanobody to bind to the RBD. This has been exploited in a sandwich ELISA and in additive neutralization experiments [23]. The figure shown is a composite of C5 (7OAO) and C1 (7OAQ (both reported [37]).

**Table 1 viruses-13-02214-t001:** Analysis of interactions at cluster 1.

Nanobody	BSA (Å^2^) ^^^	K_D_ (nM)	H/salt ^^^	EM	Xtal
V_H_H72 ^$^	766 ^&^	39	8/1 ^&^	-	6WAQ
C1	660	0.6	12/1	Disrupted	7OAP
F2	611	0.05	14/3	Disrupted	7OAY
Sb68	633	41	5/2	-	7KLW
VHH-U	650	21	10/2	-	7KN5
VHH-V	913	9	13/2	7B18, 3u, (with VHH-E)	7KN6
VHH-W	581	22	10/0	-	7KN7
NM1226	681	4	11/2	-	7NKT
Nb12 *	-	30	-	7MY3, 2u1d	-
Nb30 *	-	7	-	7MY2, 3u	-
WNb10	931	0.7	10/2	-	7LX5
SR31 ^$^	1021	6	16/0	-	7D2Z

^^^ Area of the RBD at the interface as determined by PISA [48,49]. H-bonds and salt bridges determined by Ligplot [50] and manual inspection. * EM structure of lower resolution, so comparison not valid. ^&^ The structural data are from the complex with SARS-CoV-1, not SARS-CoV-2. ^$^ Not strictly cluster 1 as it utilises an additional portion of structure.

**Table 2 viruses-13-02214-t002:** Analysis of interactions at cluster 2.

Nanobody	BSA (Å^2^) ^^^	K_D_ (nM)	H/salt ^^^	EM	Xtal
H11-H4	564	12	6/2	6ZHD, 1u2d	6ZBP
C5	739	0.1	6/4	7OAN, 3d	7OAO
H3	644	0.03	8/2	-	7OAP
mNb6 *	-	0.6	-	7KKL, 3d	-
Nb6 *	-	41	-	7KKL, 3d	-
Nb20	613	0.01	5/2	-	7JVB
NM1230	546	8	9/1	-	7B27
Sb23 *	-	11	-	7A25, 1u2d 7A29, 2u1d	-
Ty1 *	-	16	-	6ZXN, 1u2d	-
Nanosota-1	577	14	6/0		7KM5
VHH-E	775	2	3/0	7B18, 3u (with VHH-V)	7B14
WNb2	955	0.4	10/2	-	7LX5,7LDJ
MR17	903	84	10/1	-	7C8W
SR4	710	15	5/0	-	7C8V
Sb14	847	NR^&^	16/4	-	7MFU
Sb16	997	770	4/1	-	7KGK
Sb45	989	690	11/2	7N0H, 2u1d 7N0G, 1u2d	7KLW

^^^ Area of the RBD at the interface as determined by PISA [48,49]. H-bonds and salt bridges determined by Ligplot [50] and manual inspection. * EM structure of lower resolution, so comparison not valid. ^&^ Not reported.

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
