# Peer review of "Structural Biology of Nanobodies against the Spike Protein of SARS-CoV-2"

_viruses, 2021, doi:10.3390/v13112214_

Round 1

Reviewer 1 Report

The manuscript by Tang et al. provides a review on the structural details of nanobodies in complex with SARS-CoV-2 Spike protein RBD, and their biological significance. Considering that most of the work in this field has been reported within past 1.5 years and the scope of the review is highly focused, the authors wrote a rather comprehensive description of the field.

Listed below are the points that I feel need to be addressed for the manuscript to be considered fit for publication.

  1. The manuscript needs professional editing for styles, punctuation, and grammar.
  2. Line 57: “high pM” is not generally considered a low affinity. Did the authors meant “high nM”? Also, DNA shuffling is one of the methods that could be used for affinity maturation, but it is not the only method and it does not resemble the in vivo maturation process (maybe error-prone PCR more closely mimics somatic hypermutation). It is recommended that they add several other methods (e.g. CDR randomization and error-prone PCR).
  3. Lines 69-70: Many nanobodies are developed as an oligomeric form (if one can consider Fc-fused dimer as an oligomer), but there are many non-oligomeric nanobodies in clinical trials and certainly “almost always” would be an overstatement.
  4. Lines 85-86 and 90: It needs to be more clearly explained in the main text that for VHH72, the epitope does not overlap with the ACE2-binding site of RBD, but the steric hindrance by the nanobody precludes the binding of ACE2 to RBD (i.e. VHH72 competes with ACE2); and that F2 (and maybe other nanobodies) which binds to the same epitope with different orientation does not compete with ACE2.
  5. Table 1: Could the authors add the data for VHH72, if available?
  6. Are there any data available for cluster 2 nanobodies against the beta variant? If the salt bridge and cation-pi interaction with E484 is critical for the cluster 2 molecules binding to RBD, one would certainly expect greatly reduced affinity of these molecules.
  7. Line 213: Shouldn’t this be cluster 2? Also, what do RMB (line 208) and RMD (line 218) stand for?
  8. Line 224: If some cluster 1 nanobodies do not compete with ACE2, then how could they be developed as antivirals, as they would not be able to block viral infection? Is it possible that the authors did not clearly distinguish the concepts “competition” and “overlapping epitope”? (see point 4 above) “Competition” in biochemical sense primarily concerns binding kinetics and thermodynamics, and does not directly provide any molecular/structural information.
  9. Line 236: “Intra” and “inter” are not independent words, therefore “intra- or inter binding to spike trimers” needs to be rephrased, e.g. “intra- or intertrimer binding to the spike protein.”

Author Response

  1. The manuscript needs professional editing for styles, punctuation, and grammar.

The manuscript has been edited.

  1. Line 57: “high pM” is not generally considered a low affinity. Did the authors meant “high nM”? Also, DNA shuffling is one of the methods that could be used for affinity maturation, but it is not the only method and it does not resemble the in vivo maturation process (maybe error-prone PCR more closely mimics somatic hypermutation). It is recommended that they add several other methods (e.g. CDR randomization and error-prone PCR).

Changed to nM, other methods are now cited.

  1. Lines 69-70: Many nanobodies are developed as an oligomeric form (if one can consider Fc-fused dimer as an oligomer), but there are many non-oligomeric nanobodies in clinical trials and certainly “almost always” would be an overstatement.

This has been softened.

  1. Lines 85-86 and 90: It needs to be more clearly explained in the main text that for VHH72, the epitope does not overlap with the ACE2-binding site of RBD, but the steric hindrance by the nanobody precludes the binding of ACE2 to RBD (i.e. VHH72 competes with ACE2); and that F2 (and maybe other nanobodies) which binds to the same epitope with different orientation does not compete with ACE2.

This section has been rewritten to make this clearer.

  1. Table 1: Could the authors add the data for VHH72, if available?

Now added.

  1. Are there any data available for cluster 2 nanobodies against the beta variant? If the salt bridge and cation-pi interaction with E484 is critical for the cluster 2 molecules binding to RBD, one would certainly expect greatly reduced affinity of these molecules.

Yes, C5 and H3 data known, now cited.

  1. Line 213: Shouldn’t this be cluster 2? Also, what do RMB (line 208) and RMD (line 218) stand for?

Corrected to RBD

  1. Line 224: If some cluster 1 nanobodies do not compete with ACE2, then how could they be developed as antivirals, as they would not be able to block viral infection? Is it possible that the authors did not clearly distinguish the concepts “competition” and “overlapping epitope”? (see point 4 above) “Competition” in biochemical sense primarily concerns binding kinetics and thermodynamics, and does not directly provide any molecular/structural information.

The mechanism of how cluster 1 neutralise varies from agent to agent. This is now spelt out and made clear.

  1. Line 236: “Intra” and “inter” are not independent words, therefore “intra- or inter binding to spike trimers” needs to be rephrased, e.g. “intra- or intertrimer binding to the spike protein.”

This has been reworded and corrected.

Reviewer 2 Report

This focused review is an excellent effort to summarize a host of structural information concerning antibody recognition of the SARS-CoV-2 recpetor binding domain of the spike protein. The field is continually moving though efforts such as this to summarize the available data and put them in context are valuable. Though the authors found 50 references at the time of submission, there are now upwards of 100, anad there will be more.

Overall the paper is fine, though there are several places where the characterization of site of binding competition for ACE2 interaction, and inhibitory activity may be improved by specific clarification. The most confusing discussion is of the cluster 1 (class 4) nanobodies and antibodies, some of which show inhibitory and activity, and some of which do not. It would be much better if the authors clarified this up front in the Discussion of cluster 1, lines 77 and on. Indeed the way it is described, the behavior of VHH72 seems rather confusing, until one gets to details in the figure legend.

Other major conerns are that in a few places both preliminary BioRXiv postings as well as reviewed publications are mentioned, and the tabulation in the Tables cites some data from the BioRxiv rather than the reviewed publications. Specifically, refs, 30 (BioRXiv) and 51 (JBC) should be merged to refer only to the reviewed paper, and reference 48 (doi 10.1101….) and 58 should be merged.

There are a number of errors in reference to figures, and a number of grammatical inconsistencies that I note below, referenced to line numbers:

37 Figure 1 really does not illustrate binding to spike blocking ACE2 interaction.

42 specialized for specialist

57 is high pM KD a low affinity interaction?

57 define affinity in terms of KD

  1. DNA shuffling? This usually refers to exon interchanges, not to mutagenesis or block substitutions as indicated here. Further explanation would be valuable.
  2. and following. There are other classification schemes as well (Shi, Hastie). It is probably not fruitful to explicitly deal with them – but they should be referenced. The Barnes comparison should be maintained.

Figure 2a. the competition of VHH72 with ACE2 is not at all clear from this figure. There is some evidence to suggest that carbohydrate on the ACE2 provides the steric target for some of the cluster 1/class 4 Ab, Nb.

92 Does this antibody "unfold" the trimer, or stabilize an unfolded conformation?

98 However, in all complexes

137 overlapping continuum (and rewsidues of adjacent surfaces)

138 and although it is possible

141 of the collection of nanobodies

152 the text refers to Figure 3d, but the pi-cation interaction is shown in Figure 3c. A neat observation.

195 these epitopes represent two spatially distinct surface…..

196 two molecules. This was demonstrated

200 the crystal structures of F2….have been described.

203 comparison with human antibodies – this is a worthwhile comparison

209 outer face of the receptor binding motif (RBM) (or do you mean to say RBD?)

221 some that bind cluster 1 do not compete.

Figure legend VHH72, not Vhh-72

Author Response

Overall the paper is fine, though there are several places where the characterization of site of binding competition for ACE2 interaction, and inhibitory activity may be improved by specific clarification. The most confusing discussion is of the cluster 1 (class 4) nanobodies and antibodies, some of which show inhibitory and activity, and some of which do not. It would be much better if the authors clarified this up front in the Discussion of cluster 1, lines 77 and on. Indeed the way it is described, the behavior of VHH72 seems rather confusing, until one gets to details in the figure legend.

Edited and reworded to make this clearer.

Other major conerns are that in a few places both preliminary BioRXiv postings as well as reviewed publications are mentioned, and the tabulation in the Tables cites some data from the BioRxiv rather than the reviewed publications. Specifically, refs, 30 (BioRXiv) and 51 (JBC) should be merged to refer only to the reviewed paper, and reference 48 (doi 10.1101….) and 58 should be merged.

Now final papers where these are known. Citations corrected

There are a number of errors in reference to figures, and a number of grammatical inconsistencies that I note below, referenced to line numbers:

37 Figure 1 really does not illustrate binding to spike blocking ACE2 interaction.

Apologies, the citation to Figure 1 was removed

42 specialized for specialist

corrected

57 is high pM KD a low affinity interaction?

Changed to nM

57 define affinity in terms of KD

stated

58 DNA shuffling? This usually refers to exon interchanges, not to mutagenesis or block substitutions as indicated here. Further explanation would be valuable.

reworded

59 and following. There are other classification schemes as well (Shi, Hastie). It is probably not fruitful to explicitly deal with them – but they should be referenced. The Barnes comparison should be maintained.

Other classification cited.

Figure 2a. the competition of VHH72 with ACE2 is not at all clear from this figure. There is some evidence to suggest that carbohydrate on the ACE2 provides the steric target for some of the cluster 1/class 4 Ab, Nb.

Figure modified with an arrow pointing to this.

92 Does this antibody "unfold" the trimer, or stabilize an unfolded conformation?

Not known, changed to perturbed.

98 However, in all complexes

corrected

137 overlapping continuum (and residues of adjacent surfaces)

corrected

138 and although it is possible

corrected

141 of the collection of nanobodies

corrected

152 the text refers to Figure 3d, but the pi-cation interaction is shown in Figure 3c. A neat observation.

Panels 3c and 3d changed.

195 these epitopes represent two spatially distinct surface…..

corrected

196 two molecules. This was demonstrated

corrected

200 the crystal structures of F2….have been described.

corrected

203 comparison with human antibodies – this is a worthwhile comparison

Not sure how to respond

209 outer face of the receptor binding motif (RBM) (or do you mean to say RBD?)

Corrected to RBD

221 some that bind cluster 1 do not compete.

Now stated

Figure legend VHH72, not Vhh-72

corrected